# Aromatic Bromination Abolishes Deficits in Visuospatial Learning Induced by MDMA (“Ecstasy”) in Rats While Preserving the Ability to Increase LTP in the Prefrontal Cortex

**DOI:** 10.3390/ijms24043724

**Published:** 2023-02-13

**Authors:** Patricio Sáez-Briones, Boris Palma, Héctor Burgos, Rafael Barra, Alejandro Hernández

**Affiliations:** 1Laboratorio de Neurofarmacología y Comportamiento, Facultad de Ciencias Médicas, Escuela de Medicina, Universidad de Santiago de Chile, Santiago 9170022, Chile; 2Facultad de Ciencias Sociales y Humanidades, Escuela de Psicología, Universidad Autónoma de Chile, Santiago 7500912, Chile; 3Facultad de Medicina y Ciencias de la Salud, Escuela de Psicología, Universidad Mayor, Santiago 7570008, Chile; 4Centro de Investigación Biomédica y Aplicada (CIBAP), Facultad de Ciencias Médicas, Escuela de Medicina, Universidad de Santiago de Chile, Santiago 9170022, Chile; 5Laboratorio de Neurobiología, Facultad de Química y Biología, Universidad de Santiago de Chile, Santiago 9170022, Chile

**Keywords:** MDMA (3,4-methylenedioxymethamphetamine), 2-Br-4,5-MDMA, visuospatial learning, long-term potentiation, prefrontal cortex

## Abstract

It has recently been demonstrated that aromatic bromination at C(2) abolishes all typical psychomotor, and some key prosocial effects of the entactogen MDMA in rats. Nevertheless, the influence of aromatic bromination on MDMA-like effects on higher cognitive functions remains unexplored. In the present work, the effects of MDMA and its brominated analog 2Br-4,5-MDMA (1 mg/kg and 10 mg/kg i.p. each) on visuospatial learning, using a radial, octagonal Olton maze (4 × 4) which may discriminate between short-term and long-term memory, were compared with their influence on in vivo long-term potentiation (LTP) in the prefrontal cortex in rats. The results obtained indicate that MDMA diminishes both short- and long-term visuospatial memory but increases LTP. In contrast, 2Br-4,5-MDMA preserves long-term visuospatial memory and slightly accelerates the occurrence of short-term memory compared to controls, but increases LTP, like MDMA. Taken together, these data are consistent with the notion that the modulatory effects induced by the aromatic bromination of the MDMA template, which abolishes typical entactogenic-like responses, might be extended to those effects affecting higher cognitive functions, such as visuospatial learning. This effect seems not to be associated with the increase of LTP in the prefrontal cortex.

## 1. Introduction

The entactogen MDMA (3,4-methylenedioxymethamphetamine, “Ecstasy”), is a synthetic psychotropic substance capable of inducing an altered state of consciousness in humans described as a feeling of heightened self-acceptance and empathy with other persons, and reduction of negative thoughts [1]. Due to these properties, a great deal of evidence has emerged regarding potential applications of MDMA in psychotherapy [2,3,4], and as an adjunct in the treatment of neuropsychiatric disorders such as posttraumatic stress disorder [5,6], autism [7], and alcoholism [8]. Nevertheless, because of its condition as a drug of abuse, a systematic exploration of the therapeutic potential of MDMA remains controversial and incomplete [9].

The psychotropic effects of MDMA are known to be exerted by acting mainly as a special type of substrate of the serotonin transporter (SERT), thereby inducing non-exocytotic serotonin (5-HT) release by triggering a reversal of the normal transporter flux [10]. The released 5-HT, in turn, differentially stimulates several 5-HT_1_ and 5-HT_2_ receptor subtypes, thus regulating the activity of specific oxytocinergic, dopaminergic, and noradrenergic neurons, ultimately resulting in complex behavioral patterns [9]. Indeed, whereas the activation of 5-HT_1A_ and 5-HT_1B_ receptors has been proposed to promote self-confidence, together with anxiolytic-like effects, the activation of 5-HT_2A_ receptors may induce some level of narcosis. In addition, the serotonin-mediated release of oxytocin, through the activation of 5-HT_1A_ receptors, has been associated with the higher neurobehavioral features responsible for the occurrence of the entactogenic syndrome [5].

As the molecular basis of MDMA-like activity remains not fully understood, efforts have been made to identify new entactogenic-like molecules that include not only classical psychotropic phenylalkylamines [11], but also alternative structural templates, such as cathinones and benzofurans, among others [12]. In contrast, rational modifications of the benzene ring of MDMA, preserving the methylenedioxy structural moiety, remains almost unexplored. In this regard, it should be noted that aromatic halogenation usually leads to psychotropic phenylalkylamines with high in vivo potency, at least for those molecules acting as psychedelics [11]. In addition, because halogenation has been described to modulate the interactions of a drug with its molecular target by establishing so-called “halogen bonds” [13], bromination might be a useful tool to reveal some valuable hints about the modulation of MDMA activity because of its interaction with SERT [9]. Recently, we have synthesized and pharmacologically characterized, both in vitro and in vivo, a new MDMA analogue, brominated at C(2) of the aromatic ring (2-Br-4,5-MDMA), and we found that 2-Br-4,5-MDMA exhibited higher affinity for SERT than MDMA and fully abolished both platelet aggregation and ATP release, resembling the pharmacological profile of citalopram. Moreover, in vivo evaluation in rats showed that 2-Br-4,5-MDMA lacks all key MDMA-like behavioral responses in rats, including hyperlocomotion, enhanced active avoidance conditioning responses, and increased social interaction [14].

Although controversial, it has been established that MDMA may disrupt cognitive functions in humans as well as in rodents. In rats, cognitive disruption seems to be associated with alterations at the hippocampal level, possibly because of the alteration of a glutamatergic NMDA-mediated mechanism at the CA1 region [15]. In addition, repeated MDMA administration at subtoxic doses reduces visuospatial learning in rodents [16,17,18]. This evidence differs from other extensive studies that indicate the administration of sub-toxic doses of MDMA seems not to be able to induce cognitive impairment, including visuospatial learning [19].

In the present work, we set out to study, in rats, using an eight-arm radial Olton maze, if 2-Br-4,5-MDMA (Figure 1) may induce similar deficiencies in spatial memory as MDMA, or if it lacks this negative effect. In addition, we explored the ability of MDMA and 2-Br-4,5-MDMA to affect in vivo prefrontal cortex long-term potentiation (LTP)—which is instrumental for the long-standing storage of memories in the prefrontal cortex [20]—in order to find some clues as to whether the actual neuroplastic effects induced by MDMA, or 2-Br-4,5-MDMA, are associated (or not) with the modifications occurring in operative cognition.

## 2. Results 

The results obtained show that animals injected with MDMA exhibited poorer performance in the Olton radial maze compared to those receiving saline. Indeed, Figure 2A shows that the number of working memory errors decreased significantly in saline controls as the memory assay progressed in time, but not in MDMA-treated animals (intragroup statistics, ^#^
*p* < 0.05, ^###^
*p* < 0.001, two-way ANOVA followed by Bonferroni multiple comparisons test). In addition, MDMA-treated rats committed significantly more working memory errors than saline controls throughout the 12 successive daily assays (grouped in 2-day blocks) (intergroup statistics, *** *p* < 0.001, two-way ANOVA followed by Bonferroni multiple comparisons test). In striking contrast, rats injected with 2-Br-4,5-MDMA performed similarly to saline controls, both in terms of temporal decline and number of working memory errors (Figure 2A), except for the first series of sessions (blocks 1–2 and 3–4), in which the rats injected with 2-Br-4,5-MDMA committed fewer working memory errors, thus exceeding the performance of saline controls (intergroup statistics, * *p* < 0.05, ** *p* < 0.001, two-way ANOVA followed by Bonferroni multiple comparisons test).

Figure 2B shows a fairly similar picture regarding long-term memory errors, since MDMA administration resulted in a not-declining, higher amount of long-term memory errors, as compared with scores of saline controls, whereas 2-Br-4,5-MDMA-treated rats performed similar to saline controls both in terms of declining slope and amount of long-term memory errors (intragroup statistics, ^#^*p* < 0.05, ^##^
*p* < 0.01; intergroup statistics, ** *p* < 0.01, *** *p* < 0.001; two-way ANOVA followed by Bonferroni multiple comparisons test).

The time spent in solving the within-phase (working memory) and across-phase (long-term memory) tasks in the radial maze are shown in Figure 2C,D, respectively. These data again reveal that MDMA administration adversely affected both working and long-term memory, as MDMA-treated animals spent significantly more time in task solving than saline controls (intergroup statistics, *** *p* < 0.001, two-way ANOVA followed by Bonferroni multiple comparisons test). Once again, aromatic bromination at C(2) abolished the adverse effect of MDMA on short- and long-term task-solving ability in rats, (Figure 2C,D), as 2-Br-4,5-MDMA-treated rats performed similar than saline controls (two-way ANOVA followed by Bonferroni multiple comparisons test revealed no differences between 2-Br-4,5-MDMA-treated and saline-treated rats).

Figure 3A shows a scheme indicating the position of the stimulating and recording electrodes in the rat brain, as well as the averaged prefrontal cortical response evoked by contralateral stimulation of the corpus callosum in a saline control rat, prior to (basal responses, BR) and after (potentiated responses, PR) the application of the tetanizing train. Transcallosal responses evoked in the rat prefrontal cortex begin with an early downward surface positive deflection, followed by a late prominent upward surface negative wave. Detailed characterization of early and late components of these responses upon tetanization during in vivo recording, providing insight into which synapses, in which layer, are being activated during each component, has been described elsewhere [21,22,23,24]. Figure 3B shows the time course of changes in amplitude of prefrontal cortex evoked field responses prior to and after application of the potentiating train, applied 1 h after i.p. administration of either 10 mg/kg of MDMA, 1 or 10 mg/kg of 2-Br-4,5-MDMA, or saline. It can be observed that neither the drugs nor saline changed the peak-to-peak amplitude of cortical field responses prior to tetanization (between −10 and 0 min). Saline control rats exhibited about a 20% increase in peak-to-peak amplitude of cortically evoked responses after application of the tetanizing stimulation, which was maintained throughout the recording period, indicating LTP induction (intragroup statistics, * *p* < 0.05, ** *p* < 0.01, two-way ANOVA followed by Bonferroni multiple comparisons test). Intraperitoneal administration of 10 mg/kg MDMA nearly tripled the potentiating effect of the tetanizing train (implying an increase in LTP), as the peak-to-peak amplitude of cortically evoked responses increased by approximately 60% 10 min after the MDMA injection, an effect that was maintained throughout the recording session (intragroup statistics, ^#^
*p* < 0.05, ^##^
*p* < 0.01, ^###^
*p* < 0.001, two-way ANOVA followed by Bonferroni multiple comparisons test).

Injection of either 1 or 10 mg/kg 2-Br-4,5-MDMA also led to an increased potentiating effect of the stimulating train, in a dose-independent manner (Figure 3B). Figure 3C shows the overall effect of MDMA, 2-Br-4,5-MDMA, or saline in prefrontal cortex LTP induced by the potentiating train, as evaluated by the area under curves of Figure 2B. Both MDMA and 2-Br-4,5-MDMA (but not saline) similarly increased the potentiating effect of the tetanizing stimulation upon amplitude of prefrontal cortex evoked responses (intergroup statistics, * *p* < 0.05, ** *p* < 0.01, one-way ANOVA followed by Newman Keuls multiple comparisons test).

## 3. Discussion

In the present work, we assessed visuospatial learning and prefrontal cortex LTP, in vivo, in separate experiments carried out in rats inoculated with intraperitoneal 2-Br-4,5-MDMA, and compared the results against those found under MDMA. The results showed that 2-Br-4,5-MDMA completely prevented the ability of MDMA to impair visuospatial learning while maintaining the capacity of the molecule to increase prefrontal cortex LTP. These results agree with our previous findings regarding the influence of aromatic bromination of the MDMA template, as 2-Br-4,5-MDMA lacks almost all MDMA effects in vivo, highlighting the relevance of this structural modification in the alteration of the pharmacological profile of MDMA [14].

As previously mentioned, early experimental evidence has shown that administration of repeated subtoxic doses of MDMA reduces visuospatial learning in rodents [16,17,18,25]. Eventually, this effect may be induced through mechanisms including 5-HT depletion in the frontal cortex and amygdala [26], as well as the neurotoxic loss of brain 5-HT_2_ receptors [27]. In addition, MDMA may exert disruptive effects over LTP [28], while other data suggest that in vitro hippocampal LTP might be increased or diminished depending on the drug administration method (acute or chronic) [16,29,30]. A polysynaptic mechanism has been proposed to explain these differences, which includes dopaminergic D1/D5 and serotonergic 5-HT_2_ receptors [31]. As far as we are aware, no further explanations for these controversial data have been published, and a comprehensive description of the short-term and/or long-term effects of MDMA on higher cognitive functions remains incomplete. Nevertheless, it seems plausible to speculate about the possibility that an alteration in 5-HT availability may be a key factor sustaining the occurrence of learning performance. In this regard, despite of the fact that the mechanism of action of 2-Br-4,5-MDMA remains unknown, a substantial difference regarding the mode of action compared to MDMA should be expected; whereas MDMA acts as a substrate of SERT, 2-Br-4,5-MDMA seems to be able to bind to SERT in a way similar to antidepressants, such as citalopram [14]. Consequently, their differential effects on 5-HT availability might remain differential when extrapolated to memory performance, as aromatic bromination at C(2) abolishes the detrimental effects of MDMA, after chronic administration, in the Olton maze. Moreover, the presence of a bromine seems to even improve behavioral performance, at least after the first inoculations, and preserve a control-like performance for long term memory. Further research is required to extend these results in the non-toxic dose rank already known for MDMA.

Transcallosal evoked LTP in the frontal cortex is mediated by the release of glutamate from callosal axons, which acts on AMPA, NMDA, and metabotropic receptors of synapses located mainly in the dendritic spines of pyramidal cells [22,32,33,34]. Despite that, our recordings in the prefrontal cortex reflect homosynaptic plasticity (LTP in this case), meaning that the glutamatergic synapses tested for the LTP process are the same as those addressed by the tetanic stimulation protocol; cortical LTP is known to be regulated through different receptors for monoamine neurotransmitters coming from sites of the brain other than the one tetanically stimulated. This means that the MDMA-mediated increase of prefrontal cortex LTP found in the present study should be interpreted as generated by an increase in the levels of any of the neurotransmitters addressed by MDMA, namely serotonin, oxytocin, dopamine, and noradrenaline, as all these neurotransmitters can upregulate LTP in the cerebral cortex upon 5-HT_1A_ and 5-HT_2A_ [35], OXT [36], D1 [37], and β_2_ [38] receptor activation, respectively. This issue complicates the interpretation of the results because (i) there are no clear data suggesting a particular role for each of the possible receptor subtypes involved in memory processing in this cortical region, (ii) similar argumentation may apply for the prefrontal cortex LTP process, (iii) MDMA and 2-Br-4,5-MDMA were administered chronically during the learning paradigm, but only one time as a single dose short before electrophysiology, and (iv) the learning paradigm involved conscious animals, while the electrophysiological study was performed in anesthetized rats. Finally, short-term visuospatial learning is also strongly influenced by plasticity mechanisms operating in hippocampal neurons, which were not addressed in the present study.

As mentioned earlier, previous findings in rat behavioral models of spontaneous psychomotor activity showed that 2-Br-4,5-MDMA may disrupt the classical effects of MDMA on spontaneous locomotion, consistently restoring control values [14]. In contrast, the results of the present study indicate that both MDMA and 2-Br-4,5-MDMA equally increase LTP in the prefrontal cortex, implying that C(2) bromination does not affect their ability to induce neuroplastic effects in that cortical area. This dissociation between the neuroplastic effects associated with cognition (in the prefrontal cortex) and the behavioral effects associated with locomotion (presumably exerted at the level of the gait pattern generator in the spinal cord) implies that this structural modification of MDMA differentially affects the ability of this molecule to exert effects on brain regions involved in such functions. This means that at least two different mechanisms of action of MDMA would coexist, one sensitive, and the other insensitive, to bromination. Future studies of the binding of MDMA and 2-Br-4,5-MDMA to monoamine transporters may shed light on the molecular mechanisms underlying these effects.

## 4. Materials and Methods

### 4.1. Animals

The experimental protocols and animal management followed the NIH Guide for the Care and Use of Laboratory Animals [39] and were approved by the Institutional Ethics Committee of the University of Santiago de Chile (protocol 193/2020). Young, healthy male Sprague Dawley rats (180–220 g body weight) were purchased from the rearing facility of the Pontifical Catholic University of Chile and separated in shelter boxes in groups of 3–4 animals in a temperature-controlled vivarium under an inverted 12:12 h light–dark cycle (lights off from 0800 to 2000 h), with free access to standard rodent pellet diet and tap water.

### 4.2. Visuospatial Memory Performance

To separate groups of rats for the study (8 rats per group), animals were selected by a double-blind procedure and given i.p. injections of 10 mg/kg MDMA, 1 mg/kg or 10 mg/kg 2-Br-4,5-MDMA, or saline. Visuospatial memory was evaluated by employing an eight-arm radial Olton maze (from Noldus, Wageningen, the Netherlands), utilized as an Olton 4 × 4 maze according to methodology described elsewhere [40]. It consisted of eight equally spaced plexiglass arms of standard size extending from a central octagonal hub. The maze was placed on the floor level of a room with white walls. During the adaptation sessions, all arms of the maze were baited with rice puffs. Spatial cues external to the maze were provided by the experimenter themself, together with figures of different forms and tones fixed around the maze; the position of these figures and the position of the experimenter never changed during the 12 days of maze testing in each group of rats. To test animals in this maze, food motivation is required. Food motivation was induced by keeping animals on a restricted diet (8 g/day/rat) until a 10% body weight deficit was obtained (which took about 1 week). Thereafter, each animal was submitted to a 3-day adaptation period, which consisted of placing the rat in the center of the maze to explore and run to the end of the arms and consume the bait. The animals were then submitted to the visuospatial memory test (one assay daily, 12 days of testing). In each daily assay, at 10:00 h, 4 h after lights off and 30 min before beginning memory testing, rats received a single i.p. dose (1 mL/kg body weight) of either 10 mg/kg MDMA (n = 8 rats) or 10 mg/kg or 1 mg/kg 2-Br-4,5-MDMA (n = 8 rats) dissolved in saline, or saline alone (n = 8 rats). In the Olton 4 × 4 maze, each daily assay consisted of a training phase and a test phase. In the training phase, the eight arms of the maze were loaded with the bait, but four arms were blocked. The arms to be blocked were selected at random, the arrangement of blocked arms remaining constant throughout the 12 days of testing. The rats were required to enter the four unblocked arms and retrieve the bait in a period of no more than 5 min. During the subsequent test phase of each daily trial, the four blocked arms containing food were open. Rats were allowed a maximum of 5 min to retrieve the bait during the test phase. Errors were scored as entries into non baited arms during the test phase, and further subdivided into two error subtypes: (i) entry to an arm that had been entered (and the bait eaten) previously during the training phase (across-phase error) was scored as a reference memory error, this being a measure of long-term memory; (ii) re-entry into an arm from which the bait had already been retrieved during the test phase (within-phase error) was scored as a working memory error, this being a measure of short-term memory.

### 4.3. Determination of Prefrontal Cortex LTP

Experiments were carried out in three groups of six rats each, which were given i.p. injections of 10 mg/kg MDMA, 10 mg/kg or 1 mg/kg 2-Br-4,5-MDMA, or saline. Rats were weighed, anesthetized with 1.5 g/kg i.p. urethane, and placed in a stereotaxic apparatus; adequate ventilation was maintained by means of a respirator pump. LTP was induced in the rat prefrontal cortex according to a method reported elsewhere [23,24,41]. For this purpose, field cortical responses evoked by electrical stimulation of the corpus callosum (CC) were recorded by means of an electrode placed on the cortical surface (active electrode) in reference to another electrode located on the excised muscles over the frontal bone (reference electrode). Reinforcement of anesthesia during the experiments was not necessary, since surgical procedures and recordings lasted no longer than 2.5 h, and, in our experience, 1.5 g/kg i.p. urethane induces profound anesthesia lasting more than 4 h. Animals never regained consciousness and no changes in heart rate in response to stimulation were detected throughout the experiments.

After partial exposure of the frontal lobe of both cerebral hemispheres (two symmetrical holes of 2 mm diameter, each bilaterally drilled in the frontal bone), electrical stimulation of the CC was carried out by means of a bipolar electrode that penetrated through the right frontal cortex at the de Groot coordinates A = 6.8 mm, L = 2.0 mm, according to the atlas of Pellegrino and Cushman [42]. The stimulating electrode consisted of two side-by-side glued 50 µm diameter insulated tungsten wires with a 0.5 mm tip separation; one tip of the electrode was located over the CC and the other tip penetrated the CC until the de Groot coordinate V = 2.5 mm, with respect to the cortical surface. Cortical evoked responses were recorded from the left frontal cortex with a 20 µm tip diameter tungsten semi-microelectrode inserted 0.5 mm depth into the contralateral prefrontal cortex at similar surface de Groot coordinates to those utilized for transcortical stimulation of the CC. Test stimuli consisted of 100 µs duration square-wave pulses generated by means of a Grass S11 stimulator in conjunction with a Grass SIU-5 stimulus isolation unit and a Grass CCU 1A constant current unit (all Grass equipment from Astro-Med Inc., West Warwick, RI, USA). Before beginning each experiment, a full input–output series was performed at a stimulus intensity of 300–1100 µA, and test stimuli with a stimulation intensity yielding responses with peak-to-peak amplitude of 50% of the maximum were used for the remainder of the experiment. After a 30 min stabilization period, a 10 min control period of 30 averaged basal responses was recorded. Thereafter, a tetanizing stimulus consisting of a single train of 100 µs duration square-wave pulses at 312 Hz and 500 ms duration, with intensity 50% higher than the test stimuli, was applied. Recordings were amplified by a Grass P-511 preamplifier (0.8–1000 Hz bandwidth), displayed on a Philips PM 3365A digital oscilloscope, digitized at a rate of 10,000/s by an A/D converter interfaced to a computer, and stored for retrieval and off-line analysis. In all experiments body temperature and expired CO_2_ were monitored and remained within normal limits. Basal responses evoked in the rat cerebral cortex by contralateral stimulation of the CC begin with an early downward surface positive deflection (P), followed by a late upward surface negative wave (N). P–N latency and P–N peak-to-peak amplitude were measured using time and voltage cursors provided in the digital oscilloscope. Slope was determined as the amplitude/time ratio on the nearest sample to the 10% and the 90% level between cursors set on peaks P and N. The efficacy of the tetanizing train to potentiate cortical evoked responses was evaluated by measuring both the peak-to-peak amplitude and the maximal slope increase. The amplitudes were used for analyses of the experiments, according to a procedure reported elsewhere [43]. Either 10 mg/kg MDMA, 1 or 10 mg/kg 2-Br-4,5-MDMA, or saline were injected i.p. 1 h before tetanization, and changes (in percentage) of the peak-to-peak amplitude were plotted as time-course curves. To appreciate the overall effect of 2Br-4,5-MDMA and MDMA over the total period of LTP testing (1 h), the area under the curves (AUC) was determined as the integral from 0 to 60 min after tetanization using Origin 6.0 software (Microcal Software, Inc., Northampton, MA, USA) and plotted as a bar graph. At the end of the electrophysiological experiments, the animals were sacrificed with an overdose of urethane.

### 4.4. Statistical Analysis

Time course experiments (memory errors in the radial maze, LTP development, and maintenance in the prefrontal cortex) were analyzed by two-way ANOVA followed by a post hoc test for multiple comparisons. For intragroup statistical analysis of data, the Dunnett’s post hoc test was used to compare post-drug data against pre-drug baseline. For intergroup statistical analysis of data, the Bonferroni’s post hoc test was used to compare groups across different treatments. A 95% confidence interval was chosen for statistical significance. All analyses were performed using the software GraphPad Prism version 8.1.

## 5. Conclusions

Taken together, the results obtained in the present work support the notion that the modulatory effects induced by aromatic bromination at C(2) over the effects of MDMA may be extended to higher cognitive functions, such as visuospatial memory. These effects seem to be unrelated to alterations in the occurrence of LTP, at least in the prefrontal cortex. The latter indicates that 2-Br-4,5-MDMA might possess a safer pharmacological profile compared to a typical entactogen such as MDMA.

## Figures and Tables

**Figure 1 ijms-24-03724-f001:**
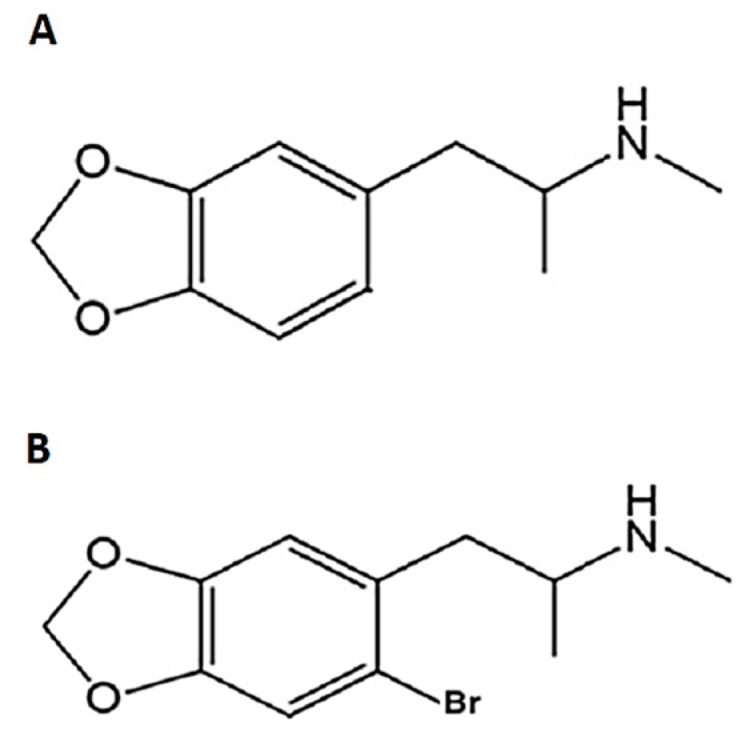
Chemical structures of (**A**) MDMA (3,4-methylenedioxymethamphetamine) and (**B**) 2Br-4,5-MDMA).

**Figure 2 ijms-24-03724-f002:**
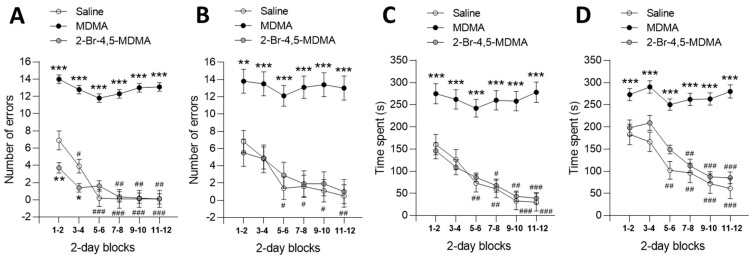
Effect of the chronic intraperitoneal administration of MDMA, 2-Br-4,5-MDMA, or saline on visuospatial memory of rats in the Olton 4 × 4 maze. Values are means ± SEM (n = 8 rats per group) of scores obtained 30 min after administration of either 10 mg/kg MDMA, 1 or 10 mg/kg 2-Br-4,5-MDMA, or saline, recorded during 12 consecutive days of testing (one assay daily) and grouped in two-assay blocks. In each block, the values were calculated as the mean of the two scores composing the block. Two-way ANOVA was used to examine visuospatial memory-related scores. Intragroup statistical analysis was performed by comparing all within-group memory error scores to that of block 1–2, using Dunnett’s multiple comparisons post hoc test (hash (#) sign indicating statistical significance). Intergroup statistical analysis was made by comparing across-group memory error scores after administration of either 10 mg/kg MDMA, 1 or 10 mg/kg 2-Br-4,5-MDMA, or saline, using Bonferroni’s multiple comparisons post hoc test (asterisks (*) sign indicating statistical significance). (**A**) Number of working memory errors: Intragroup statistics, F_(5, 126)_ = 13.64; ^#^
*p* < 0.05, ^##^
*p* < 0.01, ^###^
*p* < 0.001. Intergroup statistics, F_(2, 126)_ = 477.8; * *p* < 0.05, ** *p* < 0.01, *** *p* < 0.001. (**B**) Number of long-term memory errors: Intragroup statistics, F_(5, 126)_ = 3.722; ^#^
*p* < 0.05, ^##^
*p* < 0.01. Intergroup statistics, F_(2, 126)_ = 110.1; ** *p* < 0.01, *** *p* < 0.001. (**C**) Time spent in solving within-phase (working memory) task: Intragroup statistics, F_(5, 126)_ = 8.075; ^#^
*p* < 0.05, ^##^
*p* < 0.01, ^###^
*p* < 0.001. Intergroup statistics, F_(2, 126)_ = 173.9; *** *p* < 0.001. (**D**) Time spent in solving across-phase (long-term memory) task: Intragroup statistics, F_(5, 126)_ = 13.54; ^##^
*p* < 0.01, ^###^
*p* < 0.001. Intergroup statistics, F_(2, 126)_ = 141.5; *** *p* < 0.001.

**Figure 3 ijms-24-03724-f003:**
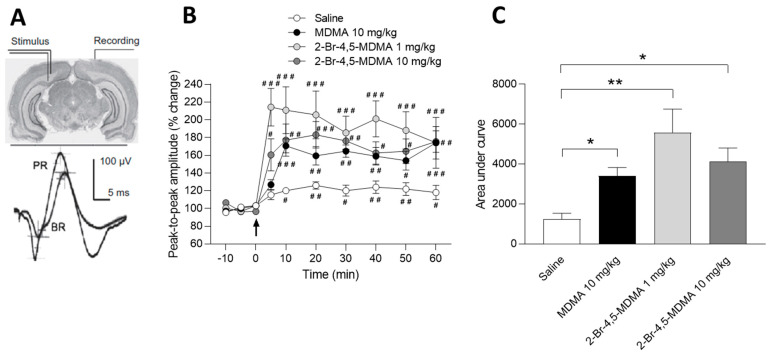
(**A**) Representative recording of the average of 12 successive basal responses (BR) and 12 successive potentiated responses (PR) evoked in the prefrontal cortex of a saline-treated control rat by contralateral stimulation of the corpus callosum at 0.1 Hz; upward deflection potential is negative. Calibration bars are shown. (**B**) Time course of the effects of either 10 mg/kg MDMA, 1 or 10 mg/kg 2-Br-4,5-MDMA, or saline, on cortical-evoked field potentials. Drugs or saline were administered i.p. 1 h before tetanization. Ordinates: change in peak-to-peak amplitude of cortical responses (percentage of three previous baseline averaged values, recorded between −10 and 0 min), after application of a single tetanizing train of electrical pulses (arrow), with intensity 50% higher than that of testing stimuli. Each point is the mean ± SEM; 30 responses were averaged per rat. Two-way ANOVA was used to examine neuroplastic changes in cortical-evoked field potentials, i.e., LTP induction. Intragroup statistical analysis allowed us to test the ability of tetanizing stimulation to develop LTP in each group, by comparing all values obtained after tetanization to the first control value recorded at min −10; hash (#) sign indicates statistical significance (F_(9, 200)_ = 22.58; ^#^
*p* < 0.05, ^##^
*p* < 0.01, ^###^
*p* < 0.001, Dunnett’s multiple comparisons post hoc test). Values are means ± SEM; n = 6 rats per group. (**C**) Global effect of either 10 mg/kg MDMA, 1 or 10 mg/kg 2-Br-4,5-MDMA, or saline on prefrontal cortex LTP, evaluated through quantification of area under corresponding time-course curves. Intergroup analysis allowed us to compare the effects of drugs and saline; asterisk (*) sign indicates statistical significance (F_(3, 20)_ = 6.171; * *p* < 0.05, ** *p* < 0.01, Newman Keuls multiple comparisons post hoc test,). Values are means ± SEM; n = 6 rats per group.

## Data Availability

Not applicable.

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
