# Peer review of "Aromatic Bromination Abolishes Deficits in Visuospatial Learning Induced by MDMA (“Ecstasy”) in Rats While Preserving the Ability to Increase LTP in the Prefrontal Cortex"

_ijms, 2023, doi:10.3390/ijms24043724_

Round 1

Reviewer 1 Report

Review on the manuscript of Sáez-Briones P et al.: “Aromatic Bromination Abolishes Deficits in Visuospatial Learning Induced by MDMA ("Ecstasy") in Rats While Preserving the Ability to Increase LTP in the Prefrontal Cortex”.

In this in vivo study, the authors explored the putative effects of the MDMA analogous 2Br-4,5-MDMA on visual-spatial learning and long-term potentiation (LTP) at the prefrontal cortex in rats. In line with what is reported in the literature, MDMA induced a reduction on the acute- and long-term visuospatial memory and increased the LTP. By contrast, although 2Br-4,5-MDMA caused a similar increase in LTP, it caused a small acceleration of acute-term visuospatial memory, but no significant effect was detected on long-term memory. Thus, the authors concluded that 2Br-4,5-MDMA was devoid of the entactogenic effects characteristic of MDMA.

The manuscript is very clear and well written. The issues that arise to me are listed below, so, I hope the authors find the following comments and suggestions useful.

1 - I recommend the authors to make a figure with the molecular structures of MDMA and 2Br-4,5-MDMA. It would help the readers to understand the difference between these 2 molecules.

2 - Mechanistically, the authors discuss that the MDMA’s effects acute- and long-term visuospatial memory may rely on 5-HT depletion in the frontal cortex and amygdala, as well as the neurotoxic loss of brain 5-HT2 receptors. Did the authors look for 5-HT levels and 5-HT2 receptors’ density? These would be straightforward experiments to mechanistically explore the differences between MDMA and 2Br-4,5-MDMA in terms of visuospatial memory. Without these experiments, the study remains very observational, not providing a mechanism for the observed effects.

Author Response

Point by point Responses to the Reviewers’ comments

REVIEWER 1.

We appreciate the constructive comments on the manuscript set out by Reviewer 1.

Review on the manuscript of Sáez-Briones P et al.: “Aromatic Bromination Abolishes Deficits in Visuospatial Learning Induced by MDMA ("Ecstasy") in Rats While Preserving the Ability to Increase LTP in the Prefrontal Cortex”.

In this in vivo study, the authors explored the putative effects of the MDMA analogous 2Br-4,5-MDMA on visual-spatial learning and long-term potentiation (LTP) at the prefrontal cortex in rats. In line with what is reported in the literature, MDMA induced a reduction on the acute- and long-term visuospatial memory and increased the LTP. By contrast, although 2Br-4,5-MDMA caused a similar increase in LTP, it caused a small acceleration of acute-term visuospatial memory, but no significant effect was detected on long-term memory. Thus, the authors concluded that 2Br-4,5-MDMA was devoid of the entactogenic effects characteristic of MDMA.

The manuscript is very clear and well written. The issues that arise to me are listed below, so, I hope the authors find the following comments and suggestions useful.

1 - I recommend the authors to make a figure with the molecular structures of MDMA and 2Br-4,5-MDMA. It would help the readers to understand the difference between these 2 molecules.

Response: Done. A figure with the molecular structures of MDMA and 2Br-4,5-MDMA is now included in the new version of the manuscript.

2 - Mechanistically, the authors discuss that the MDMA’s effects acute- and long-term visuospatial memory may rely on 5-HT depletion in the frontal cortex and amygdala, as well as the neurotoxic loss of brain 5-HT2 receptors. Did the authors look for 5-HT levels and 5-HT2 receptors’ density? These would be straightforward experiments to mechanistically explore the differences between MDMA and 2Br-4,5-MDMA in terms of visuospatial memory. Without these experiments, the study remains very observational, not providing a mechanism for the observed effects.

Response: We fully agree with this comment from reviewer 1. However, to provide a mechanistic explanation for the differential behavioral effects exerted by 2Br-4,5-MDMA versus MDMA is not an easy task for us. Aromatic bromination at C(2) of the MDMA template changes the mode of action of MDMA in SERT from a substrate into that of a citalopram-like transporter blocker, which in turn affects in rather unpredictable ways the extracellular distribution and concentration of the released serotonin. This will clearly influence the subtypes of 5-HT receptors activated in the brain by serotonin, along with the downstream involvement of other neurotransmitter receptors. To obtain a comprehensive mechanistic view of the complex effects addressed by 2Br-4,5-MDMA upon SERT targeting, extensive studies using molecular probes and pharmacological tools would be required, goals of great interest to us but beyond the goal of the present Communication.

Reviewer 2 Report

I have just one comment. I think that the introduction provides old information on MDMA, from years 2002 or 2008, this needs to be updated. I suggest putting new information in the introduction on MDMA-assisted psychoterapy in comparision to antidepressants based on Szafoni S. et al. Will MDMA-assisted psychotherapy become a breakthrough in treatment-resistant post-traumatic stress disorder? A critical narrative review. Psychiatr. Pol. 2022; 56(4): 823–836 DOI: https://doi.org/10.12740/PP/OnlineFirst/133919 - the article also contains brief history + data on how exactly MDMA works in a human body and therefore it is worth quoting and citing. The article might shed some new light as it is a really well prepared comprehensive review. Else than this, I have no other comments.

Author Response

Point by point Responses to the Reviewers’ comments

REVIEWER 2.

I have just one comment. I think that the introduction provides old information on MDMA, from years 2002 or 2008, this needs to be updated. I suggest putting new information in the introduction on MDMA-assisted psychoterapy in comparision to antidepressants based on Szafoni S. et al. Will MDMA-assisted psychotherapy become a breakthrough in treatment-resistant post-traumatic stress disorder? A critical narrative review. Psychiatr. Pol. 2022; 56(4): 823–836 DOI: https://doi.org/10.12740/PP/OnlineFirst/133919 - the article also contains brief history + data on how exactly MDMA works in a human body and therefore it is worth quoting and citing. The article might shed some new light as it is a really well prepared comprehensive review. Else than this, I have no other comments.

Response: We thank the reviewer for his comment on the need to update the possible applications of MDMA in psychotherapy, as recently stated in the recent review by Szafoni S. et al. 2022. In this regard, we included a new paragraph in the introduction summarizing the consequences of the non-exocytotic release of serotonin mediated by MDMA that sustains the entactogenic syndrome (lines 54 to 59). The review of Szafoni et al. 2022 (now reference [5]) is cited in lines 45 and 59.

Round 2

Reviewer 1 Report

Second review on the manuscript of Sáez-Briones P et al.: “Aromatic Bromination Abolishes Deficits in Visuospatial Learning Induced by MDMA ("Ecstasy") in Rats While Preserving the Ability to Increase LTP in the Prefrontal Cortex”.

Reviewer reply to authors’ response to topics 1 - the figure look very good. Congratulations for the great job.

Reviewer reply to authors’ response to topics 2 - I understand that extensive studies would be required to obtain a comprehensive mechanistic view. However, simple quantifications of 5-HT levels and 5-HT2 receptors’ density would be very helpful to make more clear conclusions.